# A delayed 400 GeV photon from GRB 221009A and implication on the intergalactic magnetic field

Zi-Qing Xia [1], Yun Wang [1], Qiang Yuan [1,2] & Yi-Zhong Fan [1,2] ✉

Large High Altitude Air Shower Observatory has detected $0.2-13$ TeV emission of GRB 221009A within 2000 s since the trigger. Here we report the detection of a 400 GeV photon, without accompanying prominent low-energy emission, by Fermi Large Area Telescope in this direction with a 0.4 days' delay. Given an intergalactic magnetic field strength of about $4 \times 10^{-17}$ G, which is comparable to limits from TeV blazars, the delayed 400 GeV photon can be explained as the cascade emission of about 10 TeV gamma rays. We estimate the probabilities of the cascade emission that can result in one detectable photon beyond 100 GeV by Fermi Large Area Telescope within $0.3-1$ days is about 2% whereas it is about 20.5% within $0.3-250$ days. Our results show that Synchrotron Self-Compton explanation is less favored with probabilities lower by a factor of about $3-30$ than the cascade scenario.

The measurement of the intergalactic magnetic field strength ($B_{\rm IGMF}$), one of the fundamental parameters of the astrophysics that may be related to how the Universe starts/evolves and carries the information of the primordial magnetic fields[1,2], is rather challenging. So far, it has not been reliably measured, yet[3–5]. One promising method, initially proposed by Plaga in ref. 6, is to explore the arrival times of gamma rays from extra-galactic TeV transients such as Gamma-ray Bursts (GRBs)[7–10] and blazars[11,12]. The basic idea is that, before reaching the observer, the primary TeV gamma rays will be absorbed by the diffuse infrared background and then generate ultra-relativistic electron-positron ($e^\pm$) pairs with Lorentz factors of $\gamma_{\rm e} \approx 9.8 \times 10^5 (1+z)(\epsilon_\gamma/1\,{\rm TeV})$, where $z$ is the redshift of the source and $\epsilon_\gamma$ is the observed energy of the primary gamma rays. These pairs will subsequently scatter off the ambient cosmic microwave background (CMB) photons, and boost them to an average energy of $\epsilon_{\gamma,2{\rm nd}} \approx 0.8\,(1+z)^2(\epsilon_\gamma/1\,{\rm TeV})^2$ GeV. Unless $B_{\rm IGMF}$ is very low (say, $\leq 10^{-20}$ G), the presence of intergalactic magnetic field will play the dominant role in delaying the arrival of the secondary gamma rays. Hence the observation of delayed gamma-ray emission can in turn impose a tight constraint or give a direct measurement of $B_{\rm IGMF}$[6,13,14].

Such an idea has been applied to the long-lasting MeV-GeV afterglow emission of GRB 940217[6,15,16] and then to other GRBs, including for instance GRB 130427A and GRB 190114C[17,18], two bursts

with powerful very high energy gamma-ray radiation[19,20]. In these studies, the limits of $B_{\rm IGMF} \geq 10^{-21} - 10^{-17}$ G have been set. Among all bursts detected so far, GRB 221009A is distinguished by its huge power (the isotropic equivalent gamma-ray radiation energy is about $10^{55}$ erg), low redshift ($z = 0.151$), and very strong TeV gamma-ray emission[21–27]. Therefore, it is an ideal target to search for the cascade emission and hence probe the intergalactic magnetic field. GRB 221009A triggered the Fermi Gamma-Ray Burst Monitor (GBM) on 2022-10-09 13:16:59 UT ($T_0$, MET 687014224), about 1 hour earlier than the Swift[21,22]. After 200 s of the Fermi-GBM trigger, the Fermi Large Area Telescope (LAT)[28] detected strong high-energy emission from GRB 221009A, and the photon flux averaged in the time interval of $T_0 + 200 - T_0 + 800$ s is about $10^{-2}$ ph cm$^{-2}$ s$^{-1}$[25]. The Large High Altitude Air Shower Observatory (LHAASO) has detected above 10 TeV emission of the extraordinary powerful GRB 221009A within about 2000 s after the trigger[27,29,30]. Though GRB 221009A is about 90 deg from the boresight at $T_0$, the Dark Matter Particle Explorer (DAMPE) also observed the significantly increasing unbiased-Trigger counts within $227-233$ s after the trigger[31].

In this work we analyze the long-term Fermi-LAT data and report the detection of a 400 GeV photon, without associated prominent low-energy emission, at approximately 0.4 days after the trigger. We show such a delayed 400 GeV photon can be explained as the cascade

[1]Key Laboratory of Dark Matter and Space Astronomy, Purple Mountain Observatory, Chinese Academy of Sciences, Nanjing 210023, China. [2]School of Astronomy and Space Science, University of Science and Technology of China, Hefei, Anhui 230026, China. ✉e-mail: yzfan@pmo.ac.cn

emission of 10 TeV photons with an intergalactic magnetic field strength of about $4 \times 10^{-17}$ G, which is more favored than Synchrotron Self-Compton (SSC) emission explanation.

## Results

### The significant detection of a delayed 400 GeV photon

In the long-term Fermi-LAT observations for GRB 221009A, we find there is a 397.7 GeV photon arriving at $T_0 + 33554$ s (approximately 0.4 days) without accompanying GeV photons (see methods subsection The Fermi-LAT data). As shown in Fig. 1, the location of this amazing event is RA = 288.252° and Dec = 19.763° given by the red filled dot, which is nicely in agreement with the Swift/Ultra-violet Optical Telescope (UVOT) localization (the blue triangle, RA = 288.265° and DEC = 19.774°[21]) as well as that of LHAASO's Water Cherenkov Detector Array (LHAASO-WCDA, the gold star, RA = 288.295° and Dec = 19.772°[29]) and Very Long Baseline Array (VLBA, the gray square, RA = 288.264° and Dec = 19.773°[32]). Pre-GRB 221009A, just two-photon events larger than 100 GeV had been found in 14 years of Fermi-LAT observations within 0.5 degree of GRB 221009A, suggesting a rather low background level at energies above 100 GeV (One was observed with the energy of 268.1 GeV at the location of RA = 288.51° and Dec = 20.08°. The other 107.1 GeV photon was located at RA = 288.47° and Dec = 19.54°). Giving its spatial and temporal coincidence with GRB 221009A, we conclude that this 400 GeV photon is indeed physically associated with this monster. We calculate the probability that this LAT event belongs to GRB 221009A within the $(T_0 + 0.3 - T_0 + 1)$ days interval using the gtsrcprob tool (in the Fermi-tools package). It turns out to be 0.9999937, corresponding to a significance level of $4.4\sigma$. Note that this 400 GeV photon is among the ULTRACLEAN class events and the possibility for being a mis-identification of a cosmic ray is very low. Therefore, we have identified the most energetic GRB photon detected by Fermi-LAT so far. The

previous records are a 95 GeV photon from GRB 130427A[19] and then a 99.3 GeV photon from GRB 221009A at an early time[23].

### An intergalactic magnetic field strength of about $4 \times 10^{-17}$ G needed in the cascade scenario

The spectral energy distributions (SEDs) measured in the time intervals after the burst of $0.05 - 0.3$ days (blue), $0.3 - 1$ days (yellow) and $0.3 - 250$ days (red) are reported in the panel (a) of Fig. 2 (see "methods" subsection The Fermi-LAT energy spectral analysis). We find that the emissions in these time intervals show different behaviors: for the $0.05 - 0.3$ days interval, it is dominated by low-energy radiation, but for the later time intervals of $0.3 - 1$ days and $0.3 - 250$ days, we only detect the single 400 GeV photon without accompanying low energy emission. In principle, the delayed GeV-TeV emission could be either from the SSC radiation of the forward shock electrons or the cascade emission of about 10 TeV prompt gamma-rays. The LHAASO collaboration has reported that the SSC afterglow from a very narrow jet, as proposed in ref. 33, can explain the TeV emission within first 2000 s[29]. With multi-band afterglow light curves of this event (see ref. 34 for more details, and an updated paper with the structured jet[35]), we obtain the expected SSC afterglow emission for the corresponding time intervals plotted as the dashed lines in panel (a) of Fig. 2. As for the $0.05 - 0.3$ days interval, we find the SEDs measured by the Fermi-LAT can be well described by this SSC model. While in the later time intervals of $0.3 - 1$ days and $0.3 - 250$ days, the fluxes measured by Fermi-LAT around 400 GeV are larger by about 3 orders of magnitude than that expected by the SSC model, which implies that it is challenging to produce such a 400 GeV photon via the SSC process. Hence, we concentrate on exploring the cascade scenario.

Within the cascade scenario, the yielding $e^\pm$ pairs have a Lorentz factor of approximately $10^7$, and the delay of the arrival time of the secondary GeV − TeV photons is governed by the deflection of the $e^\pm$ pairs by the intergalactic magnetic field. To account for a delay time of 0.4 days, we need an intergalactic magnetic field strength of approximately $4 \times 10^{-17}$ G (assuming a coherence scale of 1 Mpc), which is comparable with limits set by Fermi-LAT observations of TeV blazars[36,37] (see "methods" subsection Analytical estimate of the intergalactic magnetic field strength).

Then we conduct Monte Carlo simulation of the cascade scenario with the $B_{IGMF} = 4 \times 10^{-17}$ G to obtain the expected cascade flux (see "methods" subsection Numerical simulation of the cascade scenario and the estimate of $B_{IGMF}$.). As shown in panel (a) of Fig. 2, the expected emission flux at around 400 GeV in the cascade scenario is higher than that of the SSC model, implying that the former is more likely the origin of the photon.

## Discussion

GRB 221009A is the most powerful gamma-ray bursts detected so far. Thanks to its rather low redshift $z = 0.151$, the emission has been detected up to the energy of about 13 TeV. Because of the high optical depth of the universe to such energetic gamma rays, the intrinsic spectrum likely extends to an even higher energy range and most of these primary photons have been absorbed by the far-infrared background before reaching us. The resulting ultra-relativistic $e^\pm$ pairs will up-scatter and then boost the CMB photons to sub-TeV energy. Motivated by such a prospect, we analyze the long term of the Fermi-LAT gamma-ray observations in the direction of GRB 221009A and successfully identified a 400 GeV photon, without accompanying any low-energy gamma rays, at 0.4 days after GRB 221009A.

Motivated by the facts that the SSC afterglow model is hard to account for the data and a simple analytical estimate suggests that the cascade scenario can account for the data, we conduct the simulation of cascade with the Elmag 3.03 package and adopted the intrinsic energy spectrum of the GRB as a power-law (PL) with an index of 2.4 reported by the LHAASO collaboration[29]. In the time intervals of

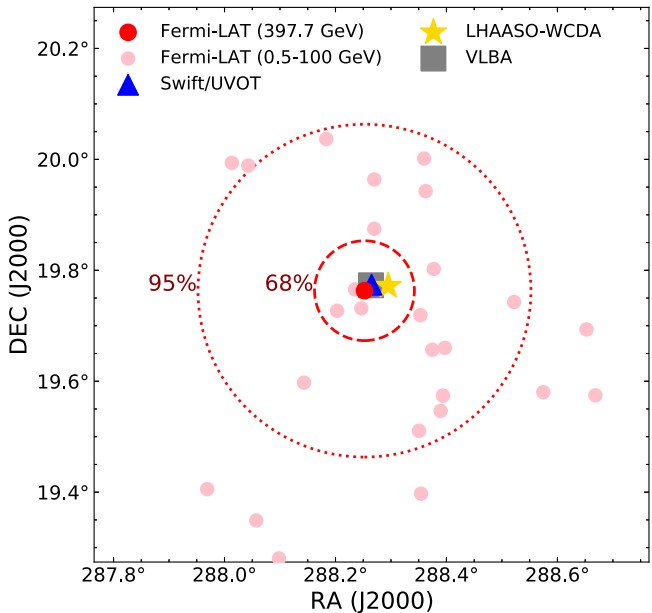

**Fig. 1 | The 1° × 1° Fermi-LAT counts map.** The map is centered at the Swift/Ultra-violet Optical Telescope (UVOT) localization (i.e., the blue triangle) for the Fermi-LAT observations in the time interval of $0.05 - 1$ days (after the burst). The red filled dot is on behalf of the delayed 397.7 GeV photon we report. The pink filled dots represent other Fermi-LAT event which is in the energy range of 0.5 GeV − 100 GeV. The gold star marks the localization of the Large High Altitude Air Shower Observatory's Water Cherenkov Detector Array (LHAASO-WCDA)[29] and the gray square represents localization of the Very Long Baseline Array (VLBA)[32]. The 68% and 95% containment angles for Fermi-LAT at 400 GeV[40] are also shown as red dashed and dotted circular lines, respectively. Source data are provided as a Source Data file.

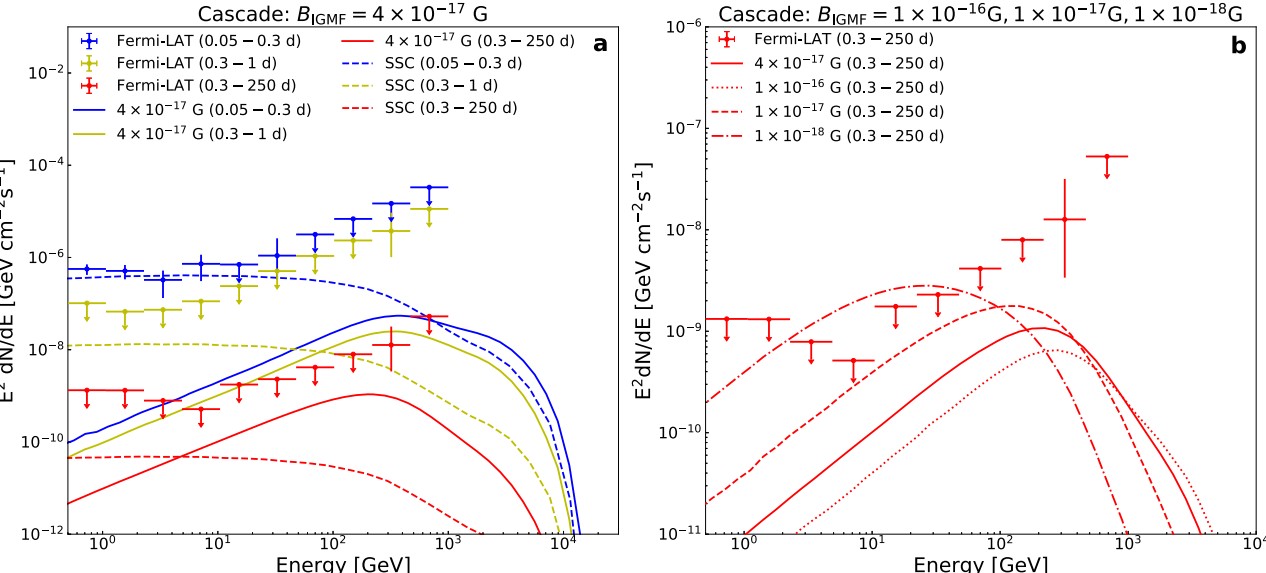

**Fig. 2 | The later-time Fermi-LAT observation of GRB 221009A and the modeling. a** The energy spectra in different time intervals (after the burst) are displayed with different color (blue: 0.05 – 0.3 days, yellow: 0.3 – 1 days, red: 0.3 – 250 days). The small filled dots with error bars or upper limits represent the measured spectral energy distributions (SEDs) in the direction of GRB 221009A. The solid lines represent the simulated spectra of cascades with the intergalactic magnetic field (IGMF) strength of $B_{IGMF} = 4 \times 10^{-17}$ G. The dashed lines represent the expected spectra of the Synchrotron Self-Compton (SSC) afterglow model. **b** We display the simulated spectra of cascades in time interval of 0.3 – 250 days with different IGMF strength of $B_{IGMF} = 1 \times 10^{-16}$ G (red dotted line), $1 \times 10^{-17}$ G (red dashed line), $1 \times 10^{-18}$ G (red dashdotted line), demonstrating the power of the long exposure data to rule out the low $B_{IGMF}$ scenario. The error bars represent the $1\sigma$ statistical uncertainties and the upper limit is at the 95% confidence level. Source data are provided as a Source Data file.

0.3 – 1 days and 0.3 – 250 days, the cascade spectra with $B_{IGMF} = 4 \times 10^{-17}$ G peak at several hundred GeV, and the corresponding probabilities that the Fermi-LAT observed one cascade photon with energy larger than 100 GeV are about 2.0% and 20.5%, respectively (see "methods" subsection Numerical simulation of the cascade scenario and the estimate of $B_{IGMF}$). Hence, we suggest that the detection of the 400 GeV photon is by chance (i.e., it is a small probability event). Anyhow, the cascade scenario for the 400 GeV photon is preferred over the SSC model with probabilities higher by a factor of 3 – 30. Although the difference is not very significant, the cascade scenario seems to be a better explanation for the delayed 400 GeV photon. Note that due to the limited observation, it is difficult to draw a strong conclusion. Considering the uncertainty of intrinsic spectrum above 13 TeV, we also take the power-law with an exponential cutoff (PLEcut) at 20 TeV model into account (see "methods" subsection Uncertainty in the intrinsic energy spectrum). The cascade fluxes above 10 GeV for the PLEcut cases are also significantly higher than that expected by the SSC model, as shown in Fig. 3. For both two intrinsic spectral models (PL and PLEcut), the possibility of a weaker IGMF $B_{IGMF} = 1 \times 10^{-18}$ G can be ruled out by the Fermi-LAT upper limits at about 10 GeV in the 0.3 – 250 days interval. A higher $B_{IGMF}$, say ≥$10^{-16}$ G, is also disfavored because of the resulting lower cascade fluxes above 100 GeV, as shown the panel (b) in Fig. 2. Therefore, a $B_{IGMF} \approx 4 \times 10^{-17}$ G seems to be preferred by the data of GRB 221009A.

The ground-based telescopes with a very large effective area, in particular the upcoming Cherenkov Telescopes Array (CTA), are expected to be able to further distinguish between different models and test such a $B_{IGMF}$ value with the observations of the powerful TeV-PeV transients and the delayed GeV-TeV emission, as shown in Fig. 4 (see "methods" subsection The prospect of testing $B_{IGMF} \approx 4 \times 10^{-17}$ G with Cherenkov Telescopes). The Very Large Gamma-ray Space Telescope (VLAST), one of the few proposed next-generation space-based GeV-TeV gamma-ray detectors with a detection area of approximately $10^5$ cm², will also be helpful in probing the IGMF in future[38].

## Methods
### The Fermi-LAT data
We focus on the long-term of Fermi-LAT gamma-ray observations above 500 MeV in the direction of GRB 221009A[39]. We select nearly 250 days ($T_0 + 0.05 - T_0 + 250$ days, MET 687018224 – 708609605) of Fermi-LAT Pass 8 R3 data[40] after the Fermi-GBM trigger in the energy range of (500 MeV – 1 TeV) within 15 degrees from the Swift/UVOT localization (RA = 288.265°, DEC = 19.774°[21]) of GRB 221009A. The FRONT + BACK conversion-type data with the SOURCE event class are adopted in our work. We exclude LAT events coming from zenith angles larger than 90° to reduce the contamination from the Earth's limb and extract good time intervals with the recommended quality-filter cuts (DATA_QUAL==1 && LAT_CONFIG==1).

### The Fermi-LAT energy spectral analysis
To perform the energy spectral analysis, we use the Fermitools package and the instrument response function P8R3_SOURCE_V3 (https://www.slac.stanford.edu/exp/glast/groups/canda/lat_Performance.htm) provided by the Fermi-LAT Collaboration. We use the make4FGLxml.py script to generate the initial model, which includes the galactic diffuse emission template (gll_iem_v07.fits), the isotropic diffuse spectral model (iso_P8R3_SOURCE_V3_v1.txt) and all the incremental Fourth Fermi-LAT source catalog[41] (gll_psc_v30.fit) sources within 25 degrees. We model the gamma-ray emission from GRB 221009A as a point source at the Swift/UVOT localization and set its spectral shape as the power-law model.

We divide the data set into three time intervals of 0.05 – 0.3 days, 0.3 – 1 days and 0.3 – 250 days after the Fermi-GBM trigger and carry out the unbinned likelihood analysis. The 0.3 – 1 days intervals is close to the arrival time of the 400 GeV photon. The selection for 0.3 – 250 days interval is intended to utilize all the information provided by the complete Fermi-LAT data up to the present. To calculate the SEDs for each time interval by separately fitting observations in 10 evenly spaced logarithmic energy bins from 500 MeV to 1 TeV. Here we fix the spectral index of GRB 221009A to 2 in each energy bin and only

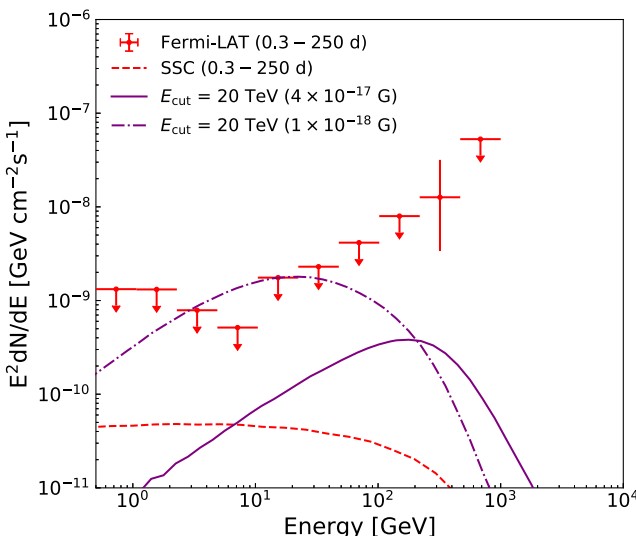

**Fig. 3 | The simulated cascade spectra for the intrinsic spectrum with an exponential energy cutoff.** Beyond 13 TeV, the intrinsic spectrum is still uncertain. For completeness, we also consider the intrinsic spectral form of power-law with an exponential energy cutoff ($E_{cut}$) of 20 TeV. The purple solid line and dashdotted line represent the simulated spectra of cascades with the intergalactic magnetic field (IGMF) strength of $B_{IGMF} = 4 \times 10^{-17}$ G and $1 \times 10^{-18}$ G, respectively. The red dashed line represents the expected spectra of the Synchrotron Self-Compton (SSC) afterglow model. The small red filled dots with error bars or upper limits represent the measured spectral energy distributions (SEDs) in the direction of GRB 221009A. The error bars represent the $1\sigma$ statistical uncertainties and the upper limit is at the 95% confidence level. These spectra are plotted in the time intervals of 0.3–250 days after the burst. Source data are provided as a Source Data file.

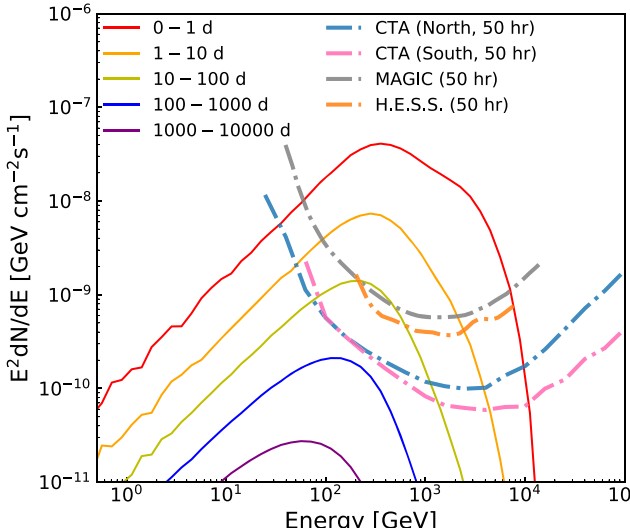

**Fig. 4 | The prospect of testing the intergalactic magnetic field (IGMF) strength $B_{IGMF} = 4 \times 10^{-17}$ G with Cherenkov Telescopes.** The simulated spectra (solid lines) of cascade emission of GRB 221009A for long time intervals with the IGMF strength $B_{IGMF} = 4 \times 10^{-17}$G and the 50-hour sensitivities (dashdotted lines) of H.E.S.S.[46], MAGIC[47], and the Cherenkov Telescopes Array (CTA[48]). The upcoming CTA can measure such emission within a few hundred days after the burst, with which the $B_{IGMF}$ can be well determined. Source data are provided as a Source Data file.

thaw the normalization of GRB 221009A and normalizations of the Galactic diffuse and isotropic diffuse components in the fitting process. The UpperLimits tool is adopted to calculate the 95% upper limit of the flux in the energy bin with a Test Statistic (TS) value < 9. in each time interval, respectively.

**Analytical estimate of the intergalactic magnetic field strength**

According to the LHAASO collaboration, gamma rays have been detected with the energy above 10 TeV[27]. At a redshift ($z$) of 0.151, the optical depth ($\tau$) of the Universe to such energetic gamma rays from interactions with photons of the intergalactic background light is quite high. Though the actual value is still not uniquely determined, it is widely believed that $\tau(z = 0.151, \epsilon_\gamma = 10 \text{ TeV}) > 5$[42], where $\epsilon_\gamma$ is the observed energy of the primary gamma rays. Consequently, most of the primary TeV gamma rays should have been absorbed, and newly generated $e^\pm$ have a Lorentz factor of $\gamma_e \approx 9.8 \times 10^6 (1+z)(\epsilon_\gamma/10 \text{ TeV})$. The created ultra-relativistic $e^\pm$ particles would Compton scatter on ambient cosmic microwave background (CMB) photons to produce high-energy secondary gamma rays. Corresponding to the highest expected number density (obtained from Eq. (6) in ref. 14), the most probable observable energy of secondary gamma rays ($\epsilon_{\gamma, 2nd}$) can be approximately estimated as

$$\epsilon_{\gamma, 2nd} \approx 265 \left(\frac{\epsilon_\gamma}{10 \text{ TeV}}\right)^2 \left(\frac{1+z}{1.151}\right)^2 \text{ GeV}, \quad (1)$$

as long as the scattering is within the Thomson regime. Note that Eq. (1) is for the most probable energy of the Inverse Compton (IC) photons and the average energy will be about 2 times smaller than this most probable energy. The average energy can be written as (4/3) $\gamma_e^2 E_{CMB} \approx 124 \left(\epsilon_\gamma/10 \text{ TeV}\right)^2 \left((1+z)/1.151\right)^2 \text{ GeV}$, where $E_{CMB}$ is the characteristic energy of the CMB photons with temperature of 2.73(1 + z) K.

Its arrival time ($t_{arr}$) is estimated to be[6,13,14]

$$t_{arr} \approx \max\{\Delta t_{TeV}, \Delta t_A, \Delta t_{IC}, \Delta t_B\}, \quad (2)$$

where $\Delta t_{TeV}$ is the observed duration of the prominent TeV emission of the source, $\Delta t_A \approx 10 (1+z)^{-1} (\epsilon_\gamma/10 \text{ TeV})^{-2} (n_{IR}/0.1 \text{cm}^{-3})^{-1}$ s is the angular spreading time delay (where $n_{IR}$ is the number density of the diffuse infrared background photons that governs the typical pair-production distance), $\Delta t_{IC} \approx 0.038 (1+z)^{-6} (\epsilon_\gamma/10 \text{ TeV})^{-3}$ s is the IC cooling time delay, and the IGMF-induced pair deflection time ($\Delta t_B$) is estimated as

$$\Delta t_B \approx 7 \times 10^5 \left(\frac{\epsilon_\gamma}{10 \text{ TeV}}\right)^{-5} \left(\frac{B_{IGMF}}{10^{-16} \text{G}}\right)^2 \left(\frac{1+z}{1.151}\right)^{-16} \text{ s}, \quad (3)$$

where $B_{IGMF}$ is the strength of IGMF and the correlation length of the magnetic field is assumed to be larger than the IC cooling radius of the pairs ($R_{IC} \approx 2\gamma_e^2 c \Delta t_{IC}/(1+z) \approx 0.1 \text{ Mpc } (1+z)^{-5}(\epsilon_\gamma/10 \text{ TeV})$).

In our analysis of the Fermi-LAT data, it is found out that at 33554 s (about 0.4 days) after the Fermi-GBM trigger there came a gamma-ray with an energy of 400 GeV. Interpreting this event as the secondary inverse Compton photon discussed above, we would have the observed energy of the primary gamma rays ($\epsilon_\gamma$) as

$$\epsilon_\gamma \approx 12 \left(\frac{\epsilon_{\gamma, 2nd}}{400 \text{ GeV}}\right)^{1/2} \left(\frac{1+z}{1.151}\right)^{-1} \text{ TeV}, \quad (4)$$

For the cascade emission of such energetic primary photons, both $\Delta t_A$ and $\Delta t_{IC}$ are negligible in comparison to $\Delta t_B$. Hence, we measure the strength of IGMF as

$$B_{IGMF} \approx 4 \times 10^{-17} \text{ G}$$
$$\times \left(\frac{\epsilon_{\gamma, 2nd}}{400 \text{ GeV}}\right)^{5/4} \left(\frac{\Delta t_B}{0.4 \text{ days}}\right)^{1/2} \left(\frac{1+z}{1.151}\right)^{11/2}. \quad (5)$$

Interestingly, this value is comparable with the lower limits set by the statistical investigation with a group of TeV blazars[36,37].

## Numerical simulation of the cascade scenario and the estimate of $B_{IGMF}$

We use the ELMAG 3.03 package to perform the Monte Carlo simulation of intergalactic electromagnetic cascades from gamma rays and electrons interacting with the extragalactic background light in the IGMF[43,44]. The IGMF is considered as a turbulent magnetic field with the root-mean-square strength of $4 \times 10^{-17}$ G and the correlation length of 1 Mpc. The model of the extragalactic background light is taken from ref. 45. We take the average of the intrinsic spectra (for standard EBL) from 200 GeV to 7 TeV within 2000 s given in Table S2 of ref. 29 as the intrinsic spectrum, which can be approximately described as a power-law (PL) with an index of about 2.4:

$$\frac{dN}{d\epsilon_\gamma} \propto \epsilon_\gamma^{-2.4}. \tag{6}$$

In this simulation, we inject 6,000,000 primary gamma rays at the redshift of $z = 0.151$ with the minimal (maximal) injection energy of 100 GeV (100 TeV). For primary photons with energies below 100 GeV, the time delay of cascade photons is larger than $10^{15}$ s according to Eq. (3), which is significantly longer than current observation time. We set the jet half-opening angle as 0.8° which was reported by ref. 29. All other parameters for the simulation are set as default values of the ELMAG 3.03 package, which are recorded in a series of input files.

Then we obtain the spectra of simulated cascades with $B_{IGMF} = 4 \times 10^{-17}$ G at arrival time intervals of $0.05 - 0.3$ days (blue), $0.3 - 1$ days (yellow) and $0.3 - 250$ days (red) after $T_0$, shown as solid lines in the panel (a) of Fig. 2. It is shown that for the $0.05 - 0.3$ days time interval, the cascade emission contributes little to the gamma-ray emission below 1 TeV. However, for the latter time interval of $0.3 - 1$ days and $0.3 - 250$ days, the cascade spectra peak at several hundred GeV, and are significantly stronger than the SSC emission above 80 GeV and 5 GeV, respectively. Especially at around 400 GeV, the flux of the cascade emission is about 7 times (2 orders of magnitude) higher than that for the SSC model in the $0.3 - 1$ days ($0.3 - 250$ days) interval. Then, we quantitatively estimate the Poisson probabilities of observing one cascade (or SSC) photon beyond 100 GeV by the Fermi-LAT with the corresponding expected number of photons in such a time interval. We multiply the cascade (or SSC) model-expected flux by the corresponding Fermi-LAT exposure in the direction of GRB 221009A (calculated by the Fermitools package) and integrate over the energy range from 100 GeV to 1 TeV to obtain the expected number of observed photons by Fermi-LAT. With $B_{IGMF} = 4 \times 10^{-17}$ G, the probabilities for the detection of one cascade photon beyond 100 GeV by the Fermi-LAT are estimated to be about 2.0% in the $0.3 - 1$ days interval and about 20.5% in the $0.3 - 250$ days interval. While in the SSC afterglow model, this probabilities are approximately 0.6% for both time intervals. So the cascade scenario is favored over the SSC model, with probabilities higher by a factor of about $3 - 30$. Moreover, we repeat the simulation with other three IGMF strengths of $B_{IGMF} = 1 \times 10^{-16}$ G, $1 \times 10^{-17}$ G, $1 \times 10^{-18}$ G. As shown in the panel (b) of Fig. 2, the possibility for smaller $B_{IGMF} = 1 \times 10^{-18}$ G can be ruled out by the Fermi-LAT upper limits of $0.3 - 250$ days interval. For $B_{IGMF} = 1 \times 10^{-16}$ G, the $> 100$ GeV cascade radiation flux will be even lower (see the panel (b) of Fig. 2), which makes the interpretation of the 400 GeV photon event more challenging. Therefore, we estimate an IGMF strength of $B_{IGMF} \approx 4 \times 10^{-17}$ G.

## Uncertainty in the intrinsic energy spectrum

Very recently, LHAASO reported the Kilometer Squared Array (KM2A) observation of GRB 221009A up to 13 TeV, which show consistency with a power-law intrinsic spectrum with a index of 2.4 for the brightest time period[30]. This is quite interesting since the energetic primary photon generating the 400 GeV cascade photon detected at about 0.4 days should have an energy of 12 TeV (see Eq. (4)), which has

indeed been recorded by LHAASO. But beyond 13 TeV, the intrinsic spectrum is still uncertain. For completeness, we also consider the intrinsic spectral form of power-law with an exponential cutoff ($E_{cut}$) of 20 TeV, labeled as PLEcut:

$$\frac{dN}{d\epsilon_\gamma} \propto \epsilon_\gamma^{-2.4} \exp\left(-\frac{\epsilon_\gamma}{E_{cut}}\right). \tag{7}$$

The simulated cascade spectra for the PLEcut models are displayed in Fig. 3 with $E_{cut} = 20$ TeV (purple) in the $0.3 - 250$ days interval, which also peak at several hundred GeV for $B_{IGMF} = 4 \times 10^{-17}$ G. Like the PL case, the cascade flux for the PLEcut models at 400 GeV is significantly higher than the SSC flux. The corresponding probability to observe one cascade photon beyond 100 GeV by the Fermi-LAT in this time interval is 9.6% for $E_{cut} = 20$ TeV, which is also significantly larger than that for the SSC model. We also see that for the PLEcut models, the Fermi-LAT data can exclude the IGMF strength $B_{IGMF} \lesssim 1 \times 10^{-18}$ G.

## The prospect of testing $B_{IGMF} \approx 4 \times 10^{-17}$ G with Cherenkov Telescopes

Observations with larger telescopes are needed to establish the cascade nature of the delayed high energy photons and then robustly pin down the $B_{IGMF}$. In Fig. 4, we compare the sensitivities (50 hours) of H.E.S.S.[46], MAGIC[47], and CTA[48] with simulated spectra for long-time cascade radiation of GRB 221009A in the case of $B_{IGMF} = 4 \times 10^{-17}$ G and the PL intrinsic spectrum. Such a signal could be well detected by H.E.S.S. and MAGIC in tens of days, and by the upcoming CTA in a few hundred days. The precise GeV − TeV spectra are able to distinguish between the cascade and SSC models (see for instance the panel (a) of Fig. 2 for the difference of the spectra). Moreover, CTA will provide a crucial test on our current evaluation of $B_{IGMF}$.

## Data availability

The data from Fermi Large Area Telescope used in this paper are publicly available via https://fermi.gsfc.nasa.gov/ssc/data/access/, obtained through the High Energy Astrophysics Science Archive Research Center (HEA-SARC) hosted by NASA's Goddard Space Flight Center. The galactic diffuse emission template (gll_iem_v07.fits) and the isotropic diffuse spectral model (iso_P8R3_SOURCE_V3_v1.txt) are available at https://fermi.gsfc.nasa.gov/ssc/data/access/lat/BackgroundModels.html. The incremental Fourth Fermi-LAT source catalog (gll_psc_v30.fit) is available at https://fermi.gsfc.nasa.gov/ssc/data/access/lat/12yr_catalog/. The datasets generated during and/or analyzed during the current study are available from the corresponding author upon request. Source data are provided with this paper.

## Code availability

The Fermitools package for analyzing the Fermi-LAT gamma-ray data are publicly available at https://fermi.gsfc.nasa.gov/ssc/data/analysis/software/. The user-contributed tool make4FGLxml.py for generating source model XML file is available at https://fermi.gsfc.nasa.gov/ssc/data/analysis/user/make4FGLxml.py. The ELMAG 3.03 package we used for the simulation of electromagnetic cascades is available at https://elmag.sourceforge.net.

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

## Acknowledgements

We acknowledge the use of the Fermi-LAT data provided by the Fermi Science Support Center. This work is supported by the National Natural Science Foundation of China No. 11921003 (Y.Z.F.), the Strategic Priority Research Program of the Chinese Academy of Sciences No. XDB0550400 (Z.Q.X and Y.W.), the National Natural Science Foundation of China Nos. 12321003 (Q.Y.), 12220101003 (Q.Y.), 12003069 (Z.Q.X.), the Project for Young Scientists in Basic Research of Chinese Academy

of Sciences No. YSBR-061 (Q.Y.), the New Cornerstone Science Foundation through the XPLORER PRIZE (Y.Z.F.), and the Entrepreneurship and Innovation Program of Jiangsu Province (Q.Y. and Z.Q.X.).

## Author contributions

Y.Z.F. launched this project. Z.Q.X. and Y.W. carried out the data analysis, following Y.Z.F. and Q.Y.'s suggestion. Y.Z.F., Q.Y., and Z.Q.X. contributed to the interpretation of data. All authors prepared the paper and joined the discussion.

## Competing interests

The authors declare no competing interests.
