## [Peer Review File · Nature Communications]

REVIEWER COMMENTS

Reviewer #1 (Remarks to the Author):

In this paper, the authors reported one ~ 400 GeV photon detected by the Fermi LAT from the brightest GRB 221009A. Using the fact that it was delayed by ~ 0.4 days from the LHAASO ~ 10 TeV photons, they reached a measurement on the poorly unknown intergalactic magnetic field strength that is $\sim 4 \times 10^{-17}$ G.

If the IG magnetic field strength can be indeed robustly measured to this value, this should be a major breakthrough in the field and the paper should be published in one of the most high-impact journals. However, with the results presented in the paper, I'm not convinced that one can make such a bold claim. Fermi LAT and LHAASO teams are preparing their own extensive analyses of a large amount of data. There are tens of other observational papers being written for this extraordinary burst. Before seeing the full picture with all the relevant data, drawing such a strong conclusion is highly risky. I understand the explorative nature of research and bold claims are encouraged. A small fraction of those bold-claim papers may turn into major discoveries but the majority will become noise and be forgotten shortly after being published. So my general suggestion to the authors is to take one of the following two options:

- Proceed with this bold claim and have it published in a regular astronomical journal and wait for confirmation or more likely disproof with the full dataset.

- Hold on the paper until seeing the full dataset (including the papers from Fermi/LAT, LHAASO, Swift-BAT/XRT teams, and many other papers from other high-energy missions). If the proposed picture is still valid and even strengthened, resubmit the paper to NC or even to the Nature main journal to report this important discovery.

In the following, I raise two major criticisms on the scientific content of the paper for the authors to consider to improve the paper whichever approach they decide to take.

1. The main argument in favor of the cascade origin of the 400 GeV photon is the steep rising spectrum at 0.3-1 days. Various IC models can also give such a rising feature. If the authors indeed want to push the measurement of the IG B field, they should spend a lot of effort ruling out various IC processes, including SSC and even EIC in case there is an X-ray-flare-like signal in the Swift XRT

lightcurve around the time when the photon was detected. This needs substantial modeling and the assessment of the unpublished Swift, Fermi, and LHAASO data.

2. The cascade model also has an important prediction about the GeV lightcurves. Given a distribution of TeV photon energies, there is a wide distribution of pair energies and, hence, a wide range of delay times for GeV photons. This would make the lightcurves smooth. Indeed, no model actually predicted an abrupt increase of GeV flux in the lightcurve. In order to make the bold claim, one needs to compare both the model lightcurves and model spectra against the full Fermi/LAT and LHAASO observations.

Reviewer #2 (Remarks to the Author):

In their paper “An inter-galactic magnetic field strength of $\sim 4 \times 10^{-17}$ G inferred with GRB 221009A”, the Authors use the publicly-available dataset resulting from the Fermi-LAT space gamma-ray telescope observations to measure the spectral energy distributions (SEDs) of gamma rays from the gamma-ray burst GRB 221009A in various time intervals spanning from 0.05 days to 50 days. In many cases, due to the limited sensitivity of Fermi-LAT, the analysis results in upper limits on the SED in specific energy bins rather than in measurements of the SED. Furthermore, the Authors identify a gamma-ray with the estimated energy of ≈ 400 GeV arriving at 33554 s after the trigger provided with the Fermi-GBM instrument. The corresponding estimated SED for the time interval of 0.3 – 1 days is $\approx 4 \times 10^{-6}$ GeV/(cm²s), while the upper limits on the SED in the energy range of 0.5 – 10 GeV in the same time interval are only $\approx (7 - 10) \times 10^{-8}$ GeV/(cm²s) (see Fig. 1). Therefore, the Authors conclude that the gamma-ray spectrum of GRB 221009A in the time interval of 0.3 – 1 days is hard, with a spectral index of $\gamma = 1.32 \pm 0.49$.

The Authors claim that “the 0.3 – 1 days emission has a hard spectrum that is inconsistent with the model of the synchrotron-self-Compton radiation of the forward shock accelerated electrons”. The Authors interpret the delayed 400 GeV gamma-ray event from GRB 221009A as a secondary gamma ray from an electromagnetic cascade developing on the intergalactic photon fields. Indeed, primary multi-TeV gamma rays absorb on the extragalactic background light, secondary electrons and positrons get deflected in the intergalactic magnetic field (IGMF); therefore, cascade gamma rays produced by these secondary electrons and positrons acquire time delay with respect to the primary gamma rays. In addition, the Authors estimate the IGMF strength $B_{\text{IGMF}} \sim 4 \times 10^{-17}$ G.

I have a number of comments and questions which I list below. At present, I am not optimistic about the prospects of publishing this paper in Nature Communications. However, I am ready to reconsider

the corrected manuscript for publication should the Editors decide that the second round of the review process is justified.

1) As was already mentioned above, the Authors estimate the gamma-ray spectral index for the time interval of 0.3 – 1 days to be $\gamma = 1.32 \pm 0.49$. Therefore, the value of the gamma-ray spectral index $\gamma_{\{2\}} = 1.32 + 0.49 = 1.81$ and even of $\gamma_{\{3\}} = 1.32 + 2 \times 0.49 = 2.3$ is possible ($\gamma_{\{2\}}$ and $\gamma_{\{3\}}$ correspond to the boundary of the one-sigma and two-sigma confidence intervals, respectively). Therefore, I strongly disagree with the claim of the Authors that “the 0.3 – 1 days emission has a hard spectrum”.

2) In addition, I strongly disagree with the claim of the Authors that “the 0.3 – 1 days (...) spectrum (...) is inconsistent with the model of the synchrotron-self-Compton radiation of the forward shock accelerated electrons”. I believe that since the value of $\gamma \approx 2$ is not excluded, it is not possible to exclude the above-mentioned model. Moreover, $\gamma \approx 1.8$ or even 1.5 could in principle be possible; such values could be achieved in blazars (for instance, see <https://ui.adsabs.harvard.edu/abs/2012A%26A...537A..47K>). Let us assume that the electrons radiating during the afterglow phase experience significant radiative losses (i.e. the loss timescale is shorter than the acceleration timescale). In this case, the spectrum of these electrons could become rather narrow, and the resulting gamma-ray spectrum could be rather hard. It is true that observations of other GRBs usually do not hint at such a feature. But these GRBs are relatively faint compared to GRB 221009A; given that GRB 221009A is so bright, there is no reason why such hard gamma-ray spectra at relatively late stages of the afterglow could not be detected.

3) Furthermore, the gamma-ray intensity at 400 GeV is severely constrained by the HAWC upper limit, as the Authors themselves show in their Fig. 3 (left). Given that circumstance, the Authors find it necessary to remark that “the probability (...) to observe one photon beyond 100 GeV by (...) Fermi-LAT in the time interval of ($T_0 + 0.3 - T_0 + 1$ days) is estimated to be $\sim 1\%$ for $B \text{ IGMF} \sim 4 \times 10^{-17} \text{ G}$. Hence, we suggest that the detection of the $\sim 400 \text{ GeV}$ photon is by chance (i.e., it is a small probability event)”. Therefore, the “true” gamma-ray intensity at 400 GeV is 2-3 orders of magnitude lower than the estimated one (based on only one 400 GeV gamma-ray), even further undermining the claim that the gamma-ray spectrum is so hard.

4) Concerning the significance of this paper to the field of gamma-ray astronomy and related fields. So far, the most reliable constraints on the IGMF strength came from blazar observations, not from gamma-ray observations (the Authors themselves seem to acknowledge that). This work does not change the trend. Therefore, the perceived impact of this paper is, at best, rather limited. Of course, I acknowledge that this conclusion is somewhat subjective.

5) The results claimed by the Authors are formulated in a rather vague manner. For instance, in the Abstract, the Authors write: “Such a hard spectrum can be generated from electromagnetic cascades initiated by early primary ~ 10 TeV photons in the intergalactic space. An inter-galactic magnetic field strength of $B_{\text{IGMF}} \sim 4 \times 10^{-17}$ G, comparable to limits from TeV blazars, can naturally account for the arrival time of the ~ 400 GeV photon as well as the HAWC non-detection.” In the Summary: “ $B_{\text{IGMF}} \sim 4 \times 10^{-17}$ G seems to be preferred by the data of GRB 221009A”. From these statements it is not clear whether the Authors truly believe that they have discovered the cascade component in the gamma-ray emission of GRB 221009A and have measured the strength of the IGMF. If so, I challenge the Authors to make direct and unequivocal statements to that effect, and to support these statements with the data analysis, statistical analysis, and theoretical interpretation. As the paper stands, I have a strong opinion that these items are still missing.

6) Finally, a rather technical comment. Due to the lack of time, I did not check some equations presented by the Authors. Should the Editors decide that the second round of the review process is justified, it could become necessary to discuss these in more details. For instance, it is not obvious whether eq. (1) is for the observable energy of secondary gamma ray or for the energy in the comoving frame (i.e. where the gamma ray was produced; for the primary gamma-ray energy of ≥ 10 TeV, in most cases, the production site is relatively near to the source). As well, it is not obvious how exactly “the most probable energy” (eq. 1) was calculated and how the “average” energy was calculated (“Note that Eq. (1) is for the most probable energy of the IC photons and the average energy can be ~ 2.5 times smaller than this most probable energy.”). The widely-known estimate $(4/3) \Gamma_e^2 E_{\text{CMB}}$ seems to be closer to the “average” energy rather than to the “most probable” energy (Γ_e here is the cascade electron Lorentz factor; E_{CMB} is the characteristic energy of the cosmic microwave background photon).

Reviewer #3 (Remarks to the Author):

The authors report a detection of 400 GeV photon 0.3-0.4 days after the prompt emission of GRB 221009A. The implied flux at this late phase is enigmatic, so that the authors provide a scenario that the photon was reprocessed one from a primary 10 TeV gamma-ray. This interpretation leads to a strength of 10^{-17} G for the inter-galactic magnetic field. The result implies that the detection of 400 GeV was by chance in spite of a very low probability. The report of the detection and interpretation for the magnetic field have significant impact worth publishing in the journal. Before accepting the paper, I'd like to provide several points to be addressed as below.

The reason to set 0.3d to divide the time-interval is not provided. This time is very close to the detection time of the 400 GeV photon. The analysis may significantly depend on the choice of this time threshold. In Fig.1, we can see a significant flux at ~ 30 GeV for $t < 0.3d$. If the detection times of

those 30 GeV photons are just before 0.3d, the flux in the time-interval 0.2d-0.4d will be high, and a hard spectrum joining the points at 30GeV and 400GeV will be obtained. In this case, the "by-chance" interpretation does not work. So we want to know the detection times of the 30 GeV photons also. Please show that the choice of the time-intervals does not affect the main results.

According to Fig.3, the HAWC upper-limit provides a strong constraint. While the authors assumed an exponential cut-off spectrum for the primary emission, the internal absorption in the emission region can lead to a much sharper cut-off in the spectrum. Or we can decrease the maximum energy or cut-off energy (adopted 10 TeV in the paper). In those cases, the spectrum of the secondary emission has a sharp cut-off above 400 GeV as well. Then, we can adopt a lower magnetic field, then the probability of the detection may be enhanced. So I think the HAWC upper-limit is not so promising. The authors should discuss the field strength neglecting the HAWC limit, whose energy range is far above the 400 GeV.

The probability of the photon detection depends on not only the model flux, but also the time-interval. Since we have only one photon at 400 GeV, it may be reasonable to widen the time-interval, e.g. 0.0-1.0d etc.

If the low-probability of the detection is unchanged by the revision, the authors should write this in abstract and main conclusion as well. This is very important information for readers.

Readers may think that the low-probability detection is awkward. The authors should discuss alternative interpretations like hadronic emission originated from high-energy protons accelerated in the GRB jet etc.

Reply to the reviewers' reports for NCOMMS-22-51642

Date: July 23, 2023

To the Reviewer #1:

In this paper, the authors reported one ~ 400 GeV photon detected by the Fermi LAT from the brightest GRB 221009A. Using the fact that it was delayed by ~ 0.4 day from the LHAASO 10 TeV photons, they reached a measurement on the poorly unknown intergalactic magnetic field strength that is $\sim 4 \times 10^{-17}$ G.

If the IG magnetic field strength can be indeed robustly measured to this value, this should be a major breakthrough in the field and the paper should be published in one of the most high-impact journals. However, with the results presented in the paper, I'm not convinced that one can make such a bold claim. Fermi LAT and LHAASO teams are preparing their own extensive analyses of a large amount of data. There are tens of other observational papers being written for this extraordinary burst. Before seeing the full picture with all the relevant data, drawing such a strong conclusion is highly risky. I understand the explorative nature of research and bold claims are encouraged. A small fraction of those bold-claim papers may turn into major discoveries but the majority will become noise and be forgotten shortly after being published. So my general suggestion to the authors is to take one of the following two options:

- Proceed with this bold claim and have it published in a regular astronomical journal and wait for confirmation or more likely disproof with the full dataset.

- Hold on the paper until seeing the full dataset (including the papers from Fermi/LAT, LHAASO, Swift-BAT/xRT teams, and many other papers from other high-energy missions). If the proposed picture is still valid and even strengthened, resubmit the paper to NC or even to the Nature main journal to report this important discovery.

Reply: Thanks for the helpful suggestion. After considering additional published data of GRB 221009A, we resubmit the substantially revised manuscript with notable improvements. In the current version, we take the recently-published LHAASO measured energy spectra as the primary TeV emission and analyze the late-time (nearly 250 days) Fermi-LAT observation. In addition, we also

exclude the possibility that the ~ 400 GeV photon originated from the SSC process.

In the following, I raise two major criticisms on the scientific content of the paper for the authors to consider to improve the paper whichever approach they decide to take.

1. The main argument in favor of the cascade origin of the 400 GeV photon is the steep rising spectrum at 0.3-1 day. Various IC models can also give such a rising feature. If the authors indeed want to push the measurement of the IG B field, they should spend a lot of effort ruling out various IC processes, including SSC and even EIC in case there is an x-ray-flare-like signal in the Swift xRT lightcurve around the time when the photon was detected. This needs substantial modeling and the assessment of the unpublished Swift, Fermi, and LHAASO data.

Reply: In the updated version, following your suggestion, we add some discussion on the IC origin for the 400 GeV photon. Among various IC models, the SSC model can explain well the measured TeV energy spectrum within first 2000 s as reported by the LHAASO collaboration (2023, Science 380, 1390–1396). Hence, here we mainly focus on the SSC model. With multi-band afterglow

Figure R1: The measured gamma-ray spectral energy distributions (SED) in the direction of GRB 221009A, the expected spectra of the SSC afterglow model and the simulated spectra of cascades in different time intervals for the IGMF strength $B_{\text{IGMF}} \sim 4 \times 10^{-17} \text{G}$.

light curves of GRB 221009A (from radio to TeV, see Ren et al. ApJ, 947, 53, 2023; an updated paper with structured jet is in preparation), we obtain the expected SSC afterglow emission for the intervals of 0.05 – 0.3 days (blue dashed line) and 0.3 – 250 days (red dashed line) as shown in Fig. R1. As for the 0.05 – 0.3 day interval, we find the SED measured by the Fermi-LAT can be well described by this SSC model. For the 0.3 – 250 day interval, the flux measured by Fermi-LAT around 400 GeV is larger by about 3 orders of magnitude than that expected by the SSC model, which implies that it’s highly impossible that such a ~ 400 GeV photon was generated by the SSC process. In addition, there is no X-ray flare found in the Swift XRT light curve at around 30000 s when the 400 GeV photon was detected¹. (Note that we have considered about 10^7 s observation of Swift XRT when calculating the expected SSC spectrum.) Compared with the predicted flux from cascade TeV photons (red solid line), the possibility that the ~ 400 GeV photon arrived at $\sim T_0 + 0.4$ days was originated from the SSC model is very unlikely.

2. The cascade model also has an important prediction about the GeV lightcurves. Given a distribution of TeV photon energies, there is a wide distribution of pair energies and, hence, a wide range of delay times for GeV photons. This would make the lightcurves smooth. Indeed, no model actually predicted an abrupt increase of GeV flux in the lightcurve. In order to make the bold claim, one needs to compare both the model lightcurves and model spectra against the full Fermi/LAT and LHAASO observations.

Reply: In the new version, we consider nearly 250 days of Fermi-LAT data (from the outburst of the GRB to recent time) and LHAASO data published recently. We take the average of the intrinsic spectra (for standard EBL) within 2000 s given in Table S2 of LHAASO’s paper (2023, Science 380, 1390–1396) as the intrinsic TeV spectrum, as shown by the green dot-dashed line in Fig. R1. The intrinsic spectrum can be approximately described as a power-law (PL) with an index of ~ 2.4 . Then we conduct the simulation of cascade with the **Elmag 3.03** package. As shown in Fig. R1, the spectrum (green solid line) of the survived primary TeV emission obtained from this simulation agrees well with the directly calculated absorbed spectrum (green dotted line). With $B_{\text{IGMF}} \sim 4 \times 10^{-17}$ G, the spectra of simulated cascades are displayed as solid lines for arrival time intervals of 0.05 – 0.3 days (blue) and 0.3 – 250 days (red) in Fig. R1. It is shown that for the 0.05 – 0.3 day time interval, the cascade emission contributes little to the gamma-ray emission below 1 TeV. However, for the 0.3 – 250 day interval, the cascade spectrum peaks at several hundred GeV, and is significantly stronger than the SSC emission above 5 GeV. Especially at around 400 GeV, the flux of the cascade emission at this time interval is higher by about two orders of magnitude than that for the SSC model. The probability that to observe one cascade photon beyond 100 GeV by the Fermi-LAT in the

¹The lightcurve of swift xRT can be found in https://www.swift.ac.uk/xrt_live_cat/01126853/

0.3 – 250 day interval is estimated to be $\sim 26.8\%$ for $B_{\text{IGMF}} \sim 4 \times 10^{-17}$ G. For the SSC afterglow model, this probability is $\sim 0.6\%$. Considering the potential uncertainty of the intrinsic spectrum above 7 TeV which is the maximum energy of LHAASO’s WCDA observation, we also take into account the PL spectrum with exponential cutoff at 10 TeV or 20 TeV (labeled as PLEcut). We find that the cascade spectra also peak at several hundred GeV for the 0.3 – 250 day time interval, as shown in Fig. 3 of the updated manuscript. Like the PL case, for both $E_{\text{cut}} = 10$ TeV and 20 TeV, the cascade fluxes for the PLEcut models at 400 GeV are significantly higher than the SSC model flux.

To the Reviewer #2:

In their paper “An inter-galactic magnetic field strength of $\sim 4 \times 10^{-17}$ G inferred with GRB 221009A”, the Authors use the publicly-available dataset resulting from the Fermi-LAT space gamma-ray telescope observations to measure the spectral energy distributions (SEDs) of gamma rays from the gamma-ray burst GRB 221009A in various time intervals spanning from 0.05 day to 50 day. In many cases, due to the limited sensitivity of Fermi-LAT, the analysis results in upper limits on the SED in specific energy bins rather than in measurements of the SED. Furthermore, the Authors identify a gamma-ray with the estimated energy of ≈ 400 GeV arriving at 33554 s after the trigger provided with the Fermi-GBM instrument. The corresponding estimated SED for the time interval of 0.3 - 1 day is $\approx 4 \times 10^{-6}$ GeV/(cm²s), while the upper limits on the SED in the energy range of 0.5 - 10 GeV in the same time interval are only $\approx (7 - 10) \times 10^{-8}$ GeV/(cm²s) (see Fig. 1). Therefore, the Authors conclude that the gamma-ray spectrum of GRB 221009A in the time interval of 0.3 - 1 day is hard, with a spectral index of $\gamma = 1.32 \pm 0.49$.

The Authors claim that “the 0.3 - 1 day emission has a hard spectrum that is inconsistent with the model of the synchrotron-self-Compton radiation of the forward shock accelerated electrons”. The Authors interpret the delayed 400 GeV gamma-ray event from GRB 221009A as a secondary gamma ray from an electromagnetic cascade developing on the intergalactic photon fields. Indeed, primary multi-TeV gamma rays absorb on the extragalactic background light, secondary electrons and positrons get deflected in the intergalactic magnetic field (IGMF); therefore, cascade gamma rays produced by these secondary electrons and positrons acquire time delay with respect to the primary gamma rays. In addition, the Authors estimate the IGMF strength $B_{\text{IGMF}} \sim 4 \times 10^{-17}$ G.

I have a number of comments and questions which I list below. At present, I am not optimistic about the prospects of publishing this paper in Nature Communications. However, I am ready to reconsider the corrected manuscript for publication should the Editors decide that the second round of the review process is justified.

Reply: Thanks for the comments and questions. With the full dataset of GRB 221009A available so far, we find our inference is still valid (even strengthened) and thus resubmit the substantially revised manuscript to Nature Communications. In this updated version, we take the recently-published LHAASO measured energy spectra (after correction for the EBL absorption) as the primary TeV emission, and analyze the late-time (nearly 250 days since the outburst of the GRB) Fermi-LAT observation. In addition, we also find that the probability of the ~ 400 GeV photon originated from the SSC process is much lower than from the cascade emission. In the following, we will respond to the comments one by one.

1) As was already mentioned above, the Authors estimate the gamma-ray spectral index for the time interval of 0.3 - 1 day to be $\gamma = 1.32 \pm 0.49$. Therefore,

the value of the gamma-ray spectral index $\gamma_2 = 1.32 + 0.49 = 1.81$ and even of $\gamma_3 = 1.32 + 2 \times 0.49 = 2.3$ is possible (γ_2 and γ_3 correspond to the boundary of the one-sigma and two-sigma confidence intervals, respectively). Therefore, I strongly disagree with the claim of the Authors that “the 0.3 - 1 day emission has a hard spectrum”.

Reply: In the revised version, we have removed such statements and directly compared the expected flux of the SSC model with that of the cascade model.

2) In addition, I strongly disagree with the claim of the Authors that “the 0.3 - 1 day (...) spectrum (...) is inconsistent with the model of the synchrotron-self-Compton radiation of the forward shock accelerated electrons”. I believe that since the value of $\gamma \approx 2$ is not excluded, it is not possible to exclude the above-mentioned model. Moreover, $\gamma \approx 1.8$ or even 1.5 could in principle be possible; such values could be achieved in blazars (for instance, see <https://ui.adsabs.harvard.edu/abs/2012A%26A...537A..47K>). Let us assume that the electrons radiating during the afterglow phase experience significant radiative losses (i.e. the loss timescale is shorter than the acceleration timescale). In this case, the spectrum of these electrons could become rather narrow, and the resulting gamma-ray spectrum could be rather hard. It is true that observations of other GRBs usually do not hint at such a feature. But these GRBs are relatively faint compared to GRB 221009A; given that GRB 221009A is so bright, there is no reason why such hard gamma-ray spectra at relatively late stages of the afterglow could not be detected.

Reply: In the revised version, following the suggestion, we add some discussion on the SSC origin for the 400 GeV photon. With multi-band afterglow light curves of GRB 221009A (from radio to TeV, see Ren et al. ApJ, 947, 53, 2023; an updated paper with structured jet is in preparation), we obtain the expected SSC afterglow emission for the 0.05–0.3 day (blue dashed line) and 0.3–250 day (red dashed line) intervals as shown in Fig. R2. As for the 0.05 – 0.3 day time interval, we find the SED measured by the Fermi-LAT can be well described by the SSC model. For the 0.3–250 day interval, the flux measured by Fermi-LAT around 400 GeV is larger by about 3 orders of magnitude than that expected by the SSC model, which implies that it’s highly impossible that such a ~ 400 GeV photon was generated from the SSC process. In addition, there is no X-ray flare found in the Swift XRT light curve (up to 10^7 s since the outburst) at around 30000 s when the 400 GeV photon was detected². Compared with the predicted flux from cascade TeV photons (red solid line), the possibility that the ~ 400 GeV photon arrived at $\sim T_0 + 0.4$ days was originated from the SSC model is very unlikely.

3) Furthermore, the gamma-ray intensity at 400 GeV is severely constrained by the HAWC upper limit, as the Authors themselves show in their Fig. 3

²The lightcurve of swift xRT can be found in <https://www.swift.ac.uk/xrt.live.cat/01126853/>

Figure R2: The measured gamma-ray spectral energy distributions (SED) in the direction of GRB 221009A, the expected spectra of the SSC afterglow model and the simulated spectra of cascades in different time intervals for the IGMF strength $B_{\text{IGMF}} \sim 4 \times 10^{-17} \text{G}$.

(left). Given that circumstance, the Authors find it necessary to remark that “the probability (...) to observe one photon beyond 100 GeV by (...) Fermi-LAT in the time interval of $(T_0 + 0.3 - T_0 + 1 \text{ day})$ is estimated to be $\sim 1\%$ for $B_{\text{IGMF}} \sim 4 \times 10^{-17} \text{G}$. Hence, we suggest that the detection of the $\sim 400 \text{ GeV}$ photon is by chance (i.e., it is a small probability event)”. Therefore, the “true” gamma-ray intensity at 400 GeV is 2 – 3 orders of magnitude lower than the estimated one (based on only one 400 GeV gamma-ray), even further undermining the claim that the gamma-ray spectrum is so hard.

Reply: Using the full dataset of GRB 221009A available so far, we have made significant revisions to the relevant parts in Sec. II and Sec. III of the manuscript:

1) The HAWC upper-limit was set by assuming a power-law spectrum with an index of 2.0 (note that this is the assumed observed spectrum rather than the intrinsic one). Generally speaking, upper-limits from air shower observations rely heavily on the assumption of spectrum. If taking different index of the power-law spectrum, the upper-limits can vary by several orders of magnitude. The LHAASO collaboration has reported the intrinsic energy spectrum of power-law with an index of 2.4 from 200 GeV to 7 TeV. The spectrum reaching the detector after the EBL absorption is even softer, and is significantly different from the assumed index of 2.0 when analyzing the HAWC data. Hence, we

expect that the HAWC upper limit is over-stringent with unrealistic assumption of spectrum and ignore it in the new version. This is also the suggestion of Reviewer 3.

2) In the new version, we consider nearly 250 days (from the outburst of the GRB to recent time) of Fermi-LAT data and LHAASO data published recently. We take the average of the intrinsic spectra (for standard EBL) within 2000 s given in Table S2 of LHAASO’s paper (2023, Science 380, 1390–1396) as the intrinsic TeV spectrum, as shown by the green dashed line in Fig. R1. The intrinsic spectrum can be approximately described as a power-law (PL) with an index of ~ 2.4 . Then we conduct the simulation of cascade with the `Elmag 3.03` package. As shown in Fig. R2, the spectrum (green solid line) of the survived primary TeV emission obtained from this simulation agrees well with the directly calculated absorbed spectrum (green dotted line). With $B_{\text{IGMF}} \sim 4 \times 10^{-17}$ G, the spectra of simulated cascades are displayed as solid lines for arrival time intervals of 0.05 – 0.3 days (blue), 0.3 – 250 days (red) in Fig. R2. It is shown that for the 0.05 – 0.3 day time interval, the cascade emission contributes little to the gamma-ray emission below 1 TeV. However, for the 0.3 – 250 day interval, the cascade spectrum peaks at several hundred GeV, and is significantly stronger than the SSC emission above 5 GeV. Especially at around 400 GeV, the flux of the cascade emission at this time interval is higher by about two orders of magnitude than that for the SSC model. The probability that to observe one cascade photon beyond 100 GeV by the Fermi-LAT in the 0.3 – 250 day interval is estimated to be $\sim 26.8\%$ for $B_{\text{IGMF}} \sim 4 \times 10^{-17}$ G. For the SSC afterglow model, this probability is $\sim 0.6\%$. Considering the potential uncertainty of the intrinsic spectrum above 7 TeV (the maximum energy of LHAASO’s WCDA observation), we also take into account the PL with exponential cutoff at 10 TeV or 20 TeV (labeled as PLEcut). We find that the cascade spectra also peak at several hundred GeV for the 0.3 – 250 day time interval, as shown in Fig. 3 of the updated version. Like the PL case, for both $E_{\text{cut}} = 10$ TeV and 20 TeV, the cascade fluxes for the PLEcut models at 400 GeV are significantly higher than the SSC model flux.

4) Concerning the significance of this paper to the field of gamma-ray astronomy and related fields. So far, the most reliable constraints on the IGMF strength came from blazar observations, not from gamma-ray observations (the Authors themselves seem to acknowledge that). This work does not change the trend. Therefore, the perceived impact of this paper is, at best, rather limited. Of course, I acknowledge that this conclusion is somewhat subjective.

Reply: We agree that the blazar observations play a very important role in probing the IGMF strength. Compared with blazar observations, GRB events have an advantage on the time distinction between the survived primary emission and the secondary cascade photons. Moreover, GRB 221009A is a very unique event (the so-called once in $10^3 - 10^4$ years) with precise TeV observations, which provides a very good opportunity to advance our knowledge. The

detection of a ~ 400 GeV photon without accompanying GeV emission suggests a hard spectrum and is less likely from the SSC process. Instead, its property is consistent with being the cascade photon. To our knowledge, this is the first time to have such a cascade photon candidate from GRBs, which is then used to infer the IGMF strength (instead of just setting limits as previous works did). This is also the first time for Fermi-LAT to convincingly detect a gamma-ray photon above 100 GeV from GRBs. We think that this work represents a significant progress in the community, and thus still pursue the publication in Nature Communications.

5) *The results claimed by the Authors are formulated in a rather vague manner. For instance, in the Abstract, the Authors write: “Such a hard spectrum can be generated from electromagnetic cascades initiated by early primary ~ 10 TeV photons in the intergalactic space. An inter-galactic magnetic field strength of $B_{\text{IGMF}} \sim 4 \times 10^{-17}$ G, comparable to limits from TeV blazars, can naturally account for the arrival time of the ~ 400 GeV photon as well as the HAWC non-detection.” In the Summary: “ $B_{\text{IGMF}} \sim 4 \times 10^{-17}$ G seems to be preferred by the data of GRB 221009A”. From these statements it is not clear whether the Authors truly believe that they have discovered the cascade component in the gamma-ray emission of GRB 221009A and have measured the strength of the IGMF. If so, I challenge the Authors to make direct and unequivocal statements to that effect, and to support these statements with the data analysis, statistical analysis, and theoretical interpretation. As the paper stands, I have a strong opinion that these items are still missing.*

Reply: As described above, we have made notable improvements (such as the more precise input of the intrinsic TeV spectrum according to LHAASO observation and the exclusion of the SSC model with multi-band observations) in the revised manuscript by using the full dataset of GRB221009A available so far. Under the current analysis, we think that the 400 GeV photons is very likely the cascade component of the very-high-energy gamma-ray emission of GRB 221009A. However, considering the fact that there is only one photon detected, we conservatively weaken the statement of the manuscript. We have clarified the relevant statements in the abstract of the new version.

6) *Finally, a rather technical comment. Due to the lack of time, I did not check some equations presented by the Authors. Should the Editors decide that the second round of the review process is justified, it could become necessary to discuss these in more details. For instance, it is not obvious whether eq. (1) is for the observable energy of secondary gamma ray or for the energy in the comoving frame (i.e. where the gamma ray was produced; for the primary gamma-ray energy of ≥ 10 TeV, in most cases, the production site is relatively near to the source). As well, it is not obvious how exactly “the most probable energy” (eq. 1) was calculated and how the “average” energy was calculated*

(“Note that Eq. (1) is for the most probable energy of the IC photons and the average energy can be ~ 2.5 times smaller than this most probable energy.”). The widely-known estimate $(4/3) \Gamma_e^2 E_{\text{CMB}}$ seems to be closer to the “average” energy rather than to the “most probable” energy (Γ_e here is the cascade electron Lorentz factor; E_{CMB} is the characteristic energy of the cosmic microwave background photon).

Reply: We agree with the Reviewer that the “average” energy of secondary gamma rays is $(4/3) \gamma_e^2 E_{\text{CMB}}$. For 10 TeV primary photon, the “average” energy can be approximately written as $\epsilon_{\gamma, \text{2nd, average}} \approx 124 \left(\frac{\epsilon_{\gamma}}{10 \text{ TeV}}\right)^2 \left(\frac{1+z}{1.151}\right)^2 \text{ GeV}$, with a Lorentz factor of $\gamma_e \approx 9.8 \times 10^6 (1+z)(\epsilon_{\gamma}/10 \text{ TeV})$. Eq. (1) in Sec. III is the approximation of the observable energy of secondary photons where the expected number density is the highest (obtained from Eq. (6) in Y.Z. Fan et al, A&A, 2004), or can be called the “most probable” energy. We have clarified it in the new version. In addition to the analytical inference, the simulation with Elmag package gives similar results, as shown in Fig. R2.

To the Reviewer #3:

The authors report a detection of 400 GeV photon 0.3-0.4 day after the prompt emission of GRB 221009A. The implied flux at this late phase is enigmatic, so that the authors provide a scenario that the photon was reprocessed one from a primary 10 TeV gamma-ray. This interpretation leads to a strength of 10^{-17} G for the inter-galactic magnetic field. The result implies that the detection of 400 GeV was by chance in spite of a very low probability. The report of the detection and interpretation for the magnetic field have significant impact worth publishing in the journal. Before accepting the paper, I'd like to provide several points to be addressed as below.

Reply: Thanks for the positive comments and recommendation. The point-to-point response to the comments is as follows.

The reason to set 0.3d to divide the time-interval is not provided. This time is very close to the detection time of the 400 GeV photon. The analysis may significantly depend on the choice of this time threshold. In Fig.1, we can see a significant flux at ~ 30 GeV for $t < 0.3d$. If the detection times of those 30 GeV photons are just before 0.3d, the flux in the time-interval 0.2d-0.4d will be high, and a hard spectrum joining the points at 30GeV and 400GeV will be obtained. In this case, the "by-chance" interpretation does not work. So we want to know the detection times of the 30 GeV photons also. Please show that the choice of the time-intervals does not affect the main results.

Reply: The 30 GeV photon arrived at 0.16 days after the trigger. For the 0.05 - 0.3 day time interval, the Fermi-LAT measured flux can be well described by the synchrotron self-Compton (SSC) model (blue dashed line), and the SSC model prediction is also higher than the cascade flux (blue solid line) throughout the Fermi-LAT energy range, as shown in Fig. R3. We therefore expect that the 30 GeV photon likely comes from the SSC process. For the 0.3 - 1 day interval (yellow lines), we see that the SSC component decreases significantly and becomes lower than the cascade component for $E > 100$ GeV. Particularly at 400 GeV, the cascade flux is higher by about one order of magnitude than the SSC component. Therefore we choose $T_0 + 0.3$ days to divide the time intervals, to reflect the cases that different component dominates the radiation.

It is true that the specific choice of time bin affects the calculation of the fluxes. In the new version, we analyze the full data (nearly 250 days, from the outburst of the GRB to recent time) of the Fermi-LAT, as plotted in red in Fig. R3. For the 0.3 - 250 day interval, the simulated cascade emission (red solid line) peaks at several hundred GeV and is significantly stronger than the SSC emission above 5 GeV. Therefore, we employ the 0.3 - 250 day interval to probe the IGMF in the new version.

According to Fig.3, the HAWC upper-limit provides a strong constraint. While the authors assumed an exponential cut-off spectrum for the primary

Figure R3: The measured gamma-ray spectral energy distributions (SED) in the direction of GRB 221009A, the expected spectra of the SSC afterglow model and the simulated spectra of cascades in different time intervals for the IGMF strength $B_{\text{IGMF}} \sim 4 \times 10^{-17} \text{G}$.

emission, the internal absorption in the emission region can lead to a much sharper cut-off in the spectrum. Or we can decrease the maximum energy or cut-off energy (adopted 10 TeV in the paper). In those cases, the spectrum of the secondary emission has a sharp cut-off above 400 GeV as well. Then, we can adopt a lower magnetic field, then the probability of the detection may be enhanced. So I think the HAWC upper-limit is not so promising. The authors should discuss the field strength neglecting the HAWC limit, whose energy range is far above the 400 GeV.

Reply: The HAWC upper-limit was set by assuming a power-law spectrum with an index of 2.0 (note that this is the assumed observed spectrum rather than the intrinsic one). Generally speaking, upper-limits from air shower observations rely heavily on the assumption of spectrum. If taking different index of the power-law spectrum, the upper-limits can vary by several orders of magnitude. The LHAASO collaboration has reported the intrinsic energy spectrum of power-law with an index of 2.4 from 200 GeV to 7 TeV. The spectrum reaching the detector after the EBL absorption is even softer, and is significantly different from the assumed index of 2.0 when analyzing the HAWC data. Hence, we expect that the HAWC upper limit is over-stringent with unrealistic assumption of spectrum and ignore it in the new version.

The impact of neglecting HAWC upper limit is minor. It does not affect

the 4×10^{-17} G estimate of the IGMF strength, but affects the conservative constraints on the IGMF if not interpreting the 400 GeV photon as the cascade emission. In the previous version, the cascade spectrum for $B_{\text{IGMF}} \sim 1 \times 10^{-17}$ G was found to be in tension with the HAWC limit. After ignoring HAWC limit, we find that the Fermi-LAT data only can exclude the IGMF strength smaller than $B_{\text{IGMF}} \sim 1 \times 10^{-18}$ G.

The probability of the photon detection depends on not only the model flux, but also the time-interval. Since we have only one photon at 400 GeV, it may be reasonable to widen the time-interval, e.g. 0.0-1.0d etc.

If the low-probability of the detection is unchanged by the revision, the authors should write this in abstract and main conclusion as well. This is very important information for readers.

Readers may think that the low-probability detection is awkward. The authors should discuss alternative interpretations like hadronic emission originated from high-energy protons accelerated in the GRB jet etc.

Reply: It is true that the width of the time bin affects the calculation of photon flux and hence the probability. In the revised version, we consider all the Fermi-LAT observation available so far from $T_0 + 0.3$ days to $T_0 + 250$ days, during which the cascade flux above 100 GeV is significantly stronger than that of the SSC model. We still find only one photon above 100 GeV from the direction of the GRB. The probability to observe one cascade photon beyond 100 GeV by the Fermi-LAT within this longer interval is estimated to be $\sim 26.8\%$ for $B_{\text{IGMF}} \sim 4 \times 10^{-17}$ G. This probability indicates that the detection of such a ~ 400 GeV photon is reasonable and credible. (The probability for the SSC model is $\sim 0.6\%$, which implies that it's highly impossible that the 400 GeV photon was generated from the SSC process compared with the cascade process.) We have added the description on the detection probability into the abstract and main conclusion of the new version.

Summary

We appreciate all Reviewers for the very helpful comments and suggestions. These comments and suggestions have been properly addressed in the revision. We believe that the current version has been significantly improved and hope that it could be found suitable for publication in Nature Communications.

Kind regards,

Zi-Qing Xia, Yun Wang, Qiang Yuan, and Yi-Zhong Fan

REVIEWER COMMENTS

Reviewer #1 (Remarks to the Author):

The authors have addressed most of my concerns and re-wrote the text in several places. In particular, they only claimed that 4×10^{-17} G is consistent with the observations rather than claiming a measurement of the IGM magnetic field. I tend to recommend the paper for publication in NC but encourage the authors to address one more comment:

It would be informative to present a 400 GeV (or in a wider band around it) light curve predicted by this model (including both the external shock component and the cascade process) and plot the existing data against it. I worry that in order to produce the abrupt increase in 400 GeV flux at 0.4 days, the earlier light curve predicted by this model may have violated the data constraint. If the authors can show that this is not an issue, I'll be happy to recommend publication.

Reviewer #2 (Remarks to the Author):

Dear Authors,

thank you very much for preparing the revised version of the manuscript.

1) However, I strongly disagree with the following conclusion made by the Authors. In their rebuttal they write:

"In the revised version, following the suggestion, we add some discussion on the SSC origin for the 400 GeV photon. With multi-band afterglow light curves of GRB 221009A (from radio to TeV, see Ren et al. ApJ, 947, 53, 2023; an updated paper with structured jet is in preparation), we obtain the expected SSC afterglow emission for the 0.05–0.3 day (blue dashed line) and 0.3–250 day (red dashed line) intervals as shown in Fig. R2. As for the 0.05 – 0.3 day time interval, we find the SED measured by the Fermi-LAT can be well described by the SSC model. For the 0.3 – 250 day interval, the flux measured by Fermi-LAT around 400 GeV is larger by about 3 orders of magnitude than that expected by the SSC model, which implies that it's highly impossible that such a ~ 400 GeV photon was generated from the SSC process."

This corresponds to their main text: (lines 72-75): "As for the 0.05 – 0.3 day interval, we find the SED measured by the Fermi-LAT can be well described by this SSC model. While in the 0.3 – 250 day

interval, the flux measured by Fermi-LAT around 400 GeV is larger by about 3 orders of magnitude than that expected by the SSC model, which implies that it's highly difficult that such a ~400 GeV photon was generated by the SSC process."

Such a result is expected since the SSC afterglow is usually expected to fade more quickly than (under certain parameters) the cascade emission. Therefore, it is not surprising that "for the 0.3 – 250 day interval, the flux measured by Fermi-LAT around 400 GeV is larger by about 3 orders of magnitude than that expected by the SSC model". This fact does not exclude the SSC model for the 400 GeV gamma-ray at all, since the value of $t_{\text{end}} = 250$ days does not have any relation to the arrival time of the 400 GeV gamma-ray, namely, 0.39 days. I believe that the reasoning of the Authors here contains a major logical flaw. I would like to suggest that the Authors take another time window, namely, a time window close in both the starting time t_{start} and end time t_{end} to the actual arrival time of the 400 GeV gamma-ray, namely, 0.39 days, and repeat their analysis. I would like to encourage the Authors to revise their manuscript accordingly.

2) "In addition, there is no X-ray flare found in the Swift XRT light curve (up to 10^7 s since the outburst) at around

30000 s when the 400 GeV photon was detected."

This is an interesting observation, but it still does not prove anything. I invite the Authors to make this point more quantitative.

3) lines 115-117: "The probability that to observe one cascade photon beyond 100 GeV by the Fermi-LAT in the 0.3 – 250 day interval is estimated to be $\sim 26.8\%$ for B IGMF $\sim 4 \times 10^{-17}$ G. While for the SSC afterglow model, this probability is $\sim 0.6\%$."

It is not clear how these probabilities were calculated. Could you please provide more details here.

4) lines 123-125: "For completeness, we also consider the intrinsic spectral form of power-law with exponential cutoff (...) with E cut = 10 TeV or 20 TeV."

Are the LHAASO data consistent with both the values of E cut = 10 TeV and 20 TeV?

Minor comment

5) lines 107-108: "We set the jet half-opening angle as 0.8° which was reported by Ref. [35]" -> half-opening angle

Reviewer #3 (Remarks to the Author):

As for my previous questions, the authors answered. But, the new analysis with 0.3-250 days interval seems ad hoc. This seems to be taken to artificially amplify the detection probability. If the other referees agree with this method, I will not disagree. But the results may be not convincing at least for me.

Second reply to the Reviewers' reports for NCOMMS-22-51642

Date: January 3, 2024

Response to the Reviewers' reports:

First of all, we would like to take this opportunity to thank all the referees for their further suggestions and comments that help us to improve our manuscript. These comments and suggestions have been properly addressed with the boldface in the updated version. The LHAASO Collaboration recently reported observations of GRB 221009A using the KM2A detector, extending the observation range to approximately 13 TeV (published in *Sci. Adv.* 9, eadj2778, 2023). It is worth noticing that the primary photon with the energy of 12 TeV (from Eq. 4), which can produce the ~ 400 GeV cascade photon arriving at ~ 0.4 day, has already been detected by LHAASO-KM2A. Please see the point-to-point reply below. We hope that the new manuscript could be found suitable for publication in *Nature Communications*.

Kind regards,

Zi-Qing Xia, Yun Wang, Qiang Yuan, and Yi-Zhong Fan

To the Reviewer #1:

The authors have addressed most of my concerns and re-wrote the text in several paces. In particular, they only claimed that $\sim 4 \times 10^{-17} G$ is consistent with the observations rather than claiming a measurement of the IGM magnetic field. I tend to recommend the paper for publication in NC but encourage the authors to address one more comment:

It would be informative to present a 400 GeV (or in a wider band around it) light curve predicted by this model (including both the external shock component and the cascade process) and plot the existing data against it. I worry that in order to produce the abrupt increase in 400 GeV flux at 0.4 days, the earlier light curve predicted by this model may have violated the data constraint. If the authors can show that this is not an issue, I'll be happy to recommend publication.

Figure R1: The light curve in one day from GRB 221009A

Reply: In Fig. R1, we plot the light curve in one day (three time intervals: 200 s - 0.05 days, 0.05 - 0.3 days, 0.3 - 1 days) after the Fermi-GBM trigger T_0 with the energy range from 218.7 GeV to 468.7 GeV corresponding to the 8th energy bin of the Fermi-LAT SED in Fig. 2. We find that for the earlier fluxes (for 200 s - 0.05 days, 0.05 - 0.3 days time intervals) predicted by this model (including both SSC and cascade processes) are below upper limits measured by Fermi-LAT and are not excluded by the observation. So we think this is not an issue.

To the Reviewer #2:

Dear Authors, thank you very much for preparing the revised version of the manuscript.

1) However, I strongly disagree with the following conclusion made by the Authors. In their rebuttal they write: “In the revised version, following the suggestion, we add some discussion on the SSC origin for the 400 GeV photon. With multi-band afterglow light curves of GRB 221009A (from radio to TeV, see Ren et al. ApJ, 947, 53, 2023; an updated paper with structured jet is in preparation), we obtain the expected SSC afterglow emission for the 0.05 - 0.3 days (blue dashed line) and 0.3 - 250 days (red dashed line) intervals as shown in Fig. R2. As for the 0.05 - 0.3 days time interval, we find the SED measured by the Fermi-LAT can be well described by the SSC model. For the 0.3 - 250 days interval, the flux measured by Fermi-LAT around 400 GeV is larger by about 3 orders of magnitude than that expected by the SSC model, which implies that it’s highly impossible that such a ~ 400 GeV photon was generated from the SSC process.”

This corresponds to their main text: (lines 72-75): “As for the 0.05 - 0.3 days interval, we find the SED measured by the Fermi-LAT can be well described by this SSC model. While in the 0.3 - 250 days interval, the flux measured by Fermi-LAT around 400 GeV is larger by about 3 orders of magnitude than that expected by the SSC model, which implies that it’s highly difficult that such a ~ 400 GeV photon was generated by the SSC process.”

Such a result is expected since the SSC afterglow is usually expected to fade more quickly than (under certain parameters) the cascade emission. Therefore, it is not surprising that “for the 0.3 - 250 days interval, the flux measured by Fermi-LAT around 400 GeV is larger by about 3 orders of magnitude than that expected by the SSC model”. This fact does not exclude the SSC model for the 400 GeV gamma-ray at all, since the value of $t_{\text{end}} = 250$ days does not have any relation to the arrival time of the 400 GeV gamma-ray, namely, 0.39 days. I believe that the reasoning of the Authors here contains a major logical flaw. I would like to suggest that the Authors take another time window, namely, a time window close in both the starting time t_{start} and end time t_{end} to the actual arrival time of the 400 GeV gamma-ray, namely, 0.39 days, and repeat their analysis. I would like to encourage the Authors to revise their manuscript accordingly.

Reply: We agree with the referee on the statement “SSC afterglow is usually expected to fade more quickly than (under certain parameters) the cascade emission.” Hence, following the suggestion, we add the analysis of the 0.3 - 1 days time interval which is close to the arrival time of the 400 GeV photon. The result is shown in the left panel Fig. 2 in the updated manuscript. At around 400 GeV, the flux of the cascade emission with intrinsic spectra reported by LHAASO at the 0.3 - 1 days interval is found to be 7 times higher than that for the SSC model. In this time interval, the corresponding probability that to observe one cascade photon beyond 100 GeV by the Fermi-LAT is estimated to

be about 2 % for $B_{\text{IGMF}} \sim 4 \times 10^{-17}$ G, which is several times larger than that for the SSC model (0.6 %). The analysis for 0.3 - 250 days time interval is mainly aimed at utilizing more complete information by the Fermi-LAT observations at later time. Furthermore, we show that the Fermi-LAT upper limits in 0.3 - 250 days interval can rule out the probability of the weaker IGMF strength $B_{\text{IGMF}} \lesssim 1 \times 10^{-18}$ G.

2) *“In addition, there is no X-ray flare found in the Swift XRT light curve (up to 10^7 s since the outburst) at around 30000 s when the 400 GeV photon was detected.” This is an interesting observation, but it still does not prove anything. I invite the Authors to make this point more quantitative.*

Reply: To obtain the best-fit SSC parameters, Ref. [37] have considered all $\sim 10^7$ s observation of Swift XRT which has included the information of no X-ray flare at around 30000 s. With these best-fit parameters, the expected SSC flux at around 400 GeV in the 0.3 - 1 days interval is 14 % of that for cascade emission. The probability for observing one SSC photon beyond 100 GeV by the Fermi-LAT in this interval is only about 0.6 %.

3) *lines 115-117: “The probability that to observe one cascade photon beyond 100 GeV by the Fermi-LAT in the 0.3 - 250 days interval is estimated to be $\sim 26\%$ for $B_{\text{IGMF}} \sim 4 \times 10^{-17}$ G. While for the SSC afterglow model, this probability is $\sim 0.6\%$.” It is not clear how these probabilities were calculated. Could you please provide more details here.*

Reply: First, we calculated the exposure of Fermi-LAT in the corresponding time interval in the direction of GRB 221009A with the `Fermitools` package. We multiplied the cascade (or SSC) model-expected flux by the obtained exposure, and then integrated over the energy range from 100 GeV to 1 TeV to calculate the expected number of observed photons by Fermi-LAT above 100 GeV. The quoted probability is hence the Poisson probability of observing one cascade (or SSC) photon beyond 100 GeV by the Fermi-LAT with the corresponding expected number of photons in such a time interval. In the updated version, we have added some detailed description on the calculation of the detection probabilities.

4) *lines 123-125: “For completeness, we also consider the intrinsic spectral form of power-law with exponential cutoff (...) with $E_{\text{cut}} = 10$ TeV or 20 TeV.” Are the LHAASO data consistent with both the values of $E_{\text{cut}} = 10$ TeV and 20 TeV?*

Reply: In the previous versions, however, only the LHAASO’s WCDA observation (up to 7 TeV) was released and made available for use. So we discussed

the uncertainty beyond 7 TeV by adding the exponential cutoff at 10 TeV or 20 TeV. Just recently, on November 15th, LHAASO released the KM2A observation of GRB 221009A and reported the detection of gamma-rays up to ~ 13 TeV (The LHAASO Collaboration, *Sci. Adv.* 9, eadj2778, 2023). They found that the intrinsic spectra from jointly fitting to the LHAASO-KM2A and LHAASO-WCDA observations are consistent with a single power-law spectrum with an index of 2.35 for the time interval of 230 s - 300 s, which corresponds to the brightest time period and contributes the majority of the initial photons. Beyond ~ 13 TeV, the intrinsic spectrum may still be uncertain. Hence, in the updated version, we retain the discussion for $E_{\text{cut}} = 20$ TeV (which has little effect on the spectrum below ~ 13 TeV) and remove that for $E_{\text{cut}} = 10$ TeV.

Minor comment 5) lines 107-108: “We set the jet half-opening angel as 0.8 degree which was reported by Ref. [35]” – “half-opening angle”.

Reply: Yes, we have corrected it.

To the Reviewer #3:

As for my previous questions, the authors answered. But, the new analysis with 0.3 - 250 days interval seems ad ho. This seems to be taken to artificially amplify the detection probability. If the other referees agree with this method, I will not disagree. But the results may be not convincing at least for me.

Reply: In order to avoid any artificial suspicion, we add the analysis for 0.3 - 1.0 days interval (close to the arrival time of the 400 GeV photon) in the updated version. In the 0.3 - 1 days interval, the cascade flux at 400 GeV is found to be 7 times higher than that for the SSC model. The detection probability of such 400 GeV photon in 0.3 - 1.0 days interval is nearly 2%, with intrinsic spectra reported by LHAASO. While for the SSC model, the probability is only 0.6 %. Even for the 0.3 - 1.0 day analysis, the cascade model is favored over the SSC model. As a comparison, the analysis with 0.3 - 250 days interval is also kept, due primarily to that we intend to include more complete Fermi-LAT observations at later time. Furthermore, the observations in 0.3 - 250 days interval can be useful to exclude the IGMF strength $B_{\text{IGMF}} \lesssim 1 \times 10^{-18} \text{G}$.

REVIEWER COMMENTS

Reviewer #1 (Remarks to the Author):

I was disappointed when seeing the lightcurve figure appended in the reply document (which is not included in the paper itself). Even though earlier model predictions are below the upper limits, the model prediction for the cascade spectrum at the time of detection is way below the detection point. This completely invalidated the suggestion that the detected photon can be used to infer the IGM magnetic field (as the title suggests). I have to conclude that this paper is not suitable for publication in NC, but may be published in a regular journal after the authors further tone down the conclusion.

Reviewer #2 (Remarks to the Author):

I am grateful to the Authors for revising their manuscript for the second time. Given that:

1) the probability "for the detection of one cascade photon beyond 100 GeV by the Fermi-LAT" is " $\sim 2.0\%$ in the 0.3 – 1 days interval" (i.e. this "probability" is quite low)

and

2) the same quantity for the "SSC afterglow model" is $\sim 0.6\%$ (i.e. not that much different from the above estimate of $\sim 2.0\%$ for the cascade model)

I feel that the claim in the abstract "Excluding its origin from the synchrotron self-Compton afterglow, we suggest that such an event is the secondary photon in electromagnetic cascades" may be a bit too strong. I invite the Authors to correct this statement if they agree with my reasoning.

However, I believe that the observation of the ~ 400 GeV gamma-ray is an interesting result. While I do not feel that the cascade origin of this event is well established, I do not oppose the publication of the paper.

Reviewer #3 (Remarks to the Author):

I think that the assumption and the analysis method are now convincing, and the quantitative result (2% detection chance) is shown. Nothing to add from my side.

Third reply to the Reviewers' reports for NCOMMS-22-51642

Date: February 21, 2024

To the Reviewer #1:

I was disappointed when seeing the lightcurve figure appended in the reply document (which is not included in the paper itself). Even though earlier model predictions are below the upper limits, the model prediction for the cascade spectrum at the time of detection is way below the detection point. This completely invalidated the suggestion that the detected photon can be used to infer the IGM magnetic field (as the title suggests). I have to conclude that this paper is not suitable for publication in NC, but may be published in a regular journal after the authors further tone down the conclusion.

Reply: Thank you very much for the further comment. We agree that neither the cascade nor the SSC process can perfectly account for the detection of the delayed 400 GeV photon, particularly for the case of small time window (0.3 – 1 days interval), as shown in our Fig. 2. Nevertheless, we have calculated the probabilities of detecting very high energy gamma rays within these two models and found out that the cascade scenario is favored over the SSC model (the detection of a 400 GeV cascade gamma ray by Fermi-LAT has probabilities of 2.0% within 0.3 – 1 days and 20.5% within 0.3 – 250 days, while the probabilities for the SSC model are only about 0.6%, lower by a factor of $\sim 3 - 30$ than the cascade scenario). Though people do need bigger instrument like CTA to observe similar events with larger statistics to firmly establish the cascade origin and then robustly infer the intergalactic magnetic field strength, we think that the current work represents a significant progress on this topic and deserves the publication in Nature Communications for the following reasons:

1) As a record-breaking event distinguished by its huge amount of isotropic equivalent energy and the strong early TeV radiation, GRB 221009A has attracted wide attention. Here, for the *first* time we report the detection of a 400 GeV photon arriving at ~ 0.4 days after the burst. To our knowledge, this is the most energetic photon detected by Fermi-LAT from GRBs. The previous record was the 99.3 GeV photon detected in the prompt emission phase of GRB 221009A. The detection of a ~ 400 GeV photon, with a delay time of 0.4 days, without associated prominent low energy emission by a space telescope with an effective area of $\sim 10^4$ cm² is surprising and intriguing.

2) The early high energy emission of GRB 221009A has been well measured up to ~ 13 TeV, and such energetic photons should inevitably suffer from serious absorption before reaching us. There should be a cascade radiation component though its arrival time is sensitively dependent on the strength of the intergalactic magnetic field and is thus uncertain. The typical energy of the secondary cascade gamma rays is expected to be a few hundred GeV. Though with a single photon (its detection, given the small effective area of the space telescope, is likely by chance), it is challenging to conclusively pin down its origin, the cascade scenario is an attractive possibility and deserves a careful exploration. Interestingly, the probability of the cascade origin is found to be higher than the SSC scenario. We think this is very encouraging.

In the revision, we have softened some claims in our manuscript by modifying the statements from the conclusive language to the potential explanation. The revisions include the title, abstract, the main text and conclusion, which can properly present our main finding and the theoretical interpretation. The uncertainties have been extensively discussed and the prospect of further tests is projected. Hopefully this improved version could be found suitable for publication in Nature Communications.

To the Reviewer #2:

I am grateful to the Authors for revising their manuscript for the second time. Given that: 1) the probability “for the detection of one cascade photon beyond 100 GeV by the Fermi–LAT” is “ $\sim 2.0\%$ in the 0.3 – 1 days interval” (i.e. this “probability” is quite low) and 2) the same quantity for the “SSC afterglow model” is $\sim 0.6\%$ (i.e. not that much different from the above estimate of $\sim 2.0\%$ for the cascade model) I feel that the claim in the abstract “Excluding its origin from the synchrotron self–Compton afterglow, we suggest that such an event is the secondary photon in electromagnetic cascades” may be a bit too strong. I invite the Authors to correct this statement if they agree with my reasoning.

However, I believe that the observation of the ~ 400 GeV gamma ray is an interesting result. While I do not feel that the cascade origin of this event is well established, I do not oppose the publication of the paper.

Reply: We sincerely appreciate your favorable recommendation on our manuscript. Following your suggestion, we have softened some claims in our manuscript by modifying the statements from the conclusive language to the potential explanation. In particular, we have revised the title, the abstract, some parts of the main text and the conclusion to more properly present the main finding and the theoretical interpretation.

To the Reviewer #3:

I think that the assumption and the analysis method are now convincing, and the quantitative result (2% detection chance) is shown. Nothing to add from my side.

Reply: Thanks a lot for your favorable comments.

REVIEWERS' COMMENTS

Reviewer #2 (Remarks to the Author):

I am grateful to the Authors for revising their manuscript. The changes at this round were rather minor. I believe that the revised version of the text, indeed, describes the results obtained by the Authors better than the "old" version of the text. The Authors find that "the cascade scenario" [for the ~ 400 GeV gamma-ray] "is preferred over the SSC model with possibilities higher by a factor of 3 – 30" (probably it is worth to replace "possibilities" by "probabilities" or something similar here). This difference (by 3 – 30 times) is not that big; nevertheless, let me repeat that I find the observation of the ~ 400 GeV gamma-ray to be an interesting result.

Again, I do not oppose the publication of the paper. As was speculated by me at the first round of the review, the initial promise of the "discovery" (i.e. a clear, say, $>5 \sigma$ statistical significance preference for the cascade model), indeed, did not materialise. Due to the scarcity of the dataset, this was expected. I leave the final decision to the Editors of the Journal. If the paper gets accepted, it would probably benefit from a moderate degree of language editing. I believe that could be done at the processing stage.

Fourth reply to the Reviewer' reports for NCOMMS-22-51642

Date: April 15, 2024

To the Reviewer #2:

I am grateful to the Authors for revising their manuscript. The changes at this round were rather minor. I believe that the revised version of the text, indeed, describes the results obtained by the Authors better than the “old” version of the text. The Authors find that “the cascade scenario” [for the ~ 400 GeV gamma-ray] “is preferred over the SSC model with possibilities higher by a factor of 3 - 30” (probably it is worth to replace “possibilities” by “probabilities” or something similar here). This difference (by 3 - 30 times) is not that big; nevertheless, let me repeat that I find the observation of the ~ 400 GeV gamma-ray to be an interesting result.

Reply: Thank you very much for your further suggestions and approval of our text revision. We have replaced “possibilities” by “probabilities” in the corresponding text. We also appreciate your interest on the detection of the delayed 400 GeV photon.

Again, I do not oppose the publication of the paper. As was speculated by me at the first round of the review, the initial promise of the “discovery” (i.e. a clear, say, $\geq 5\sigma$ statistical significance preference for the cascade model), indeed, did not materialise. Due to the scarcity of the dataset, this was expected. I leave the final decision to the Editors of the Journal. If the paper gets accepted, it would probably benefit from a moderate degree of language editing. I believe that could be done at the processing stage.

Reply: As you mentioned, indeed, the observation are limited and it is difficult to draw a strong conclusion. We have further illustrated this fact in “Discussion” section and softened some corresponding text.